# Infrared and Visible Image Fusion Based on Visual Saliency Map and Image Contrast Enhancement

**DOI:** 10.3390/s22176390

**Published:** 2022-08-25

**Authors:** Yuanyuan Liu, Zhiyong Wu, Xizhen Han, Qiang Sun, Jian Zhao, Jianzhuo Liu

**Affiliations:** 1Changchun Institute of Optics, Fine Mechanics and Physics, Chinese Academy of Sciences, Changchun 130000, China; 2University of Chinese Academy of Sciences, Beijing 100049, China; 3Computer Science and Technology College, Changchun University of Science and Technology, Changchun 130000, China; 4Suzhou Institute of Biomedical Engineering and Technology, Chinese Academy of Sciences, Suzhou 215000, China

**Keywords:** infrared and visible images, multi-scale decomposition, image fusion, contrast enhancement, visual saliency map

## Abstract

The purpose of infrared and visible image fusion is to generate images with prominent targets and rich information which provides the basis for target detection and recognition. Among the existing image fusion methods, the traditional method is easy to produce artifacts, and the information of the visible target and texture details are not fully preserved, especially for the image fusion under dark scenes and smoke conditions. Therefore, an infrared and visible image fusion method is proposed based on visual saliency image and image contrast enhancement processing. Aiming at the problem that low image contrast brings difficulty to fusion, an improved gamma correction and local mean method is used to enhance the input image contrast. To suppress artifacts that are prone to occur in the process of image fusion, a differential rolling guidance filter (DRGF) method is adopted to decompose the input image into the basic layer and the detail layer. Compared with the traditional multi-scale decomposition method, this method can retain specific edge information and reduce the occurrence of artifacts. In order to solve the problem that the salient object of the fused image is not prominent and the texture detail information is not fully preserved, the salient map extraction method is used to extract the infrared image salient map to guide the fusion image target weight, and on the other hand, it is used to control the fusion weight of the basic layer to improve the shortcomings of the traditional ‘average’ fusion method to weaken the contrast information. In addition, a method based on pixel intensity and gradient is proposed to fuse the detail layer and retain the edge and detail information to the greatest extent. Experimental results show that the proposed method is superior to other fusion algorithms in both subjective and objective aspects.

## 1. Introduction

The main purpose of image fusion is to generate a single image containing complementary information from multiple images of the same scene. In the field of multi-sensor image fusion, infrared and visible image fusion is an important technology. Since infrared sensors can capture thermal information about targets in a scene, some objects can be highlighted in weak light, occlusion and bad weather conditions. The visible image contains the texture details of the scene, and the images presented are more consistent with human visual perception in terms of intensity and contrast. These two types of images have complementary effects and their fusion can obtain more accurate and richer scene information. At present, this technology plays an important role in security monitoring, target detection, target recognition and so on [1,2,3,4,5].

Ma et al. investigated the field of infrared and visible image fusion [6]. The fusion algorithm can be divided into seven categories according to the theoretical way: multi-scale transform, sparse representation, neural network, subspace, dominance-based, hybrid model and other methods. Among them, the multi-scale transform algorithm is most widely used typically adopting Laplace pyramid transform [7] (LP), wavelet transform [8] (Haar), non-subsampled shearlet transform [9] (NSST) or non-subsampled contourlet transform [10] (NSCT). These algorithms are based on the Laplace pyramid method, which smoothes the image to a certain extent and loses some structure and detail information. At the same time, it may also produce artifacts that affect the quality of the fused image.

These problems have also attracted the attention of many researchers. In order to retain more detail information and structure information, some researchers proposed a method to optimize the objective function by imposing constraint conditions to achieve fusion. Huang et al. [11] reserved detail information based on gradient constraints and explicit constraints, highlighting the explicit goal. Li et al. [12] used contrast fidelity and sparse constraints for image fusion. There are also improved fusion methods based on multi-scale decomposition Chen et al. [13] proposed a target enhanced multi-scale transformation (MST) decomposition model by using linear programming to decompose the source image and fuse them on different scales, spatial resolution and decomposition layers. This method introduces a linear transformation function to control the weight of infrared images and fully retains the information of infrared prominent targets. Li et al. [14] proposed a multi-scale fusion method of potential low-rank representation decomposition to extract global and local structural information from the source image, and fully tap the information of the source image. These methods focus on preserving image information and ignore the problem of artifacts caused by the loss of image information in multi-scale decomposition. Ding et al. [15] proposed a fusion method based on non-subsampled shearlet transform and sparse structural features. The sparsity based on structural information is used to guide the fusion of high-frequency coefficients, and the principal component analysis is used to fuse low-frequency information. This method has a good effect on extracting prominent target information. However, due to the abandonment of some low-frequency information in the principal component analysis, some information will be lost or artifacts will occur.

In recent years, the edge-preserving filter has been successfully applied to image fusion. Zhou et al. [16] proposed a multi-scale decomposition method based on Gaussian and bilateral filtering mixing, but this result will also smooth the contrast information. Ma et al. [17] proposed a multi-scale decomposition method based on the combination of Gaussian and rolling guided filter (*RGF*), which has a certain effect on artifact elimination. In recent years, the method based on explicitness has gradually emerged, and more people are keen to use the method of combining the multi-scale method with explicitness to fuse images. Zuo et al. [18] proposed an infrared and visible image fusion method that introduced region segmentation into the dual-tree complex wavelet transform (DTCWT) region. According to explicitness, the region of interest and the background region were separated, and the results were mapped to the fusion results.

Although information retention and artifacts elimination in image fusion are individually studied in the above methods, they have not been considered comprehensively at the same time. Based on the advantages and disadvantages of the above methods, this paper proposes an infrared and visible image fusion method based on a visual saliency map and image contrast enhancement on the basis of two typical scenes that affect the fusion effect, namely, dark scenes and smoke background. The advantages of the algorithm in this paper are as follows. In view of the difficulty caused by the low image contrast to the fusion, we propose an improved method of gamma correction and local mean to enhance the contrast of the input image. In view of the problem that artifacts are prone to occur in the process of image fusion, a DRGF method is adopted to decompose the input image into the basic layer and the detail layer. Compared with the traditional multi-scale decomposition method, this method can retain specific edge information and reduce the occurrence of artifacts. In order to solve the problem that the visible target of the fused image is not prominent and the texture detail information is not fully preserved, a method is proposed to extract the visible image. On the one hand, it is used to extract the visible image from the infrared image to guide the fusion image target weight. On the other hand, it is used to control the fusion weight of the basic layer to improve the shortcomings of the traditional ‘average’ fusion method to weaken the contrast information. In addition, a method based on pixel intensity and gradient is proposed to fuse the detail layer and retain the edge and detail information to the greatest extent.

## 2. Methods

The structure of the fusion algorithm is shown in Figure 1. Firstly, the contrast of the input image is enhanced by an improved gamma correction and local mean. Then, the infrared image is extracted to guide the target weight of the fusion image. Then, the input image is decomposed into the basic layer and the detail layer by using the DRGF multi-scale decomposition method. For the base layer, the fusion weight is controlled by using the dominant value. For the detail layer, the fusion method based on pixel intensity and gradient is used.

### 2.1. Image Contrast Enhancement

Low contrast images in a dark background will have a certain impact on the fusion quality [19]. Considering this aspect, this paper adds the image contrast enhancement algorithm to the fusion framework. The purpose of image contrast enhancement is to improve the visual effect of the image and enhance the contrast and clarity, which can be divided into spatial domain enhancement and frequency domain enhancement. The spatial domain enhancement algorithms include histogram equalization (HE), gamma transform, and logarithmic image processing model (LIP). The frequency domain enhancement methods include high-pass filtering, low-pass filtering and homomorphic filtering. Gamma transform [20] is mainly used for image correction, and the images with too high and too low gray levels are corrected accordingly to achieve the contrast enhancement effect. The simple form of this method is expressed as Equation (Equation 1).
(1)$$P=L^{\gamma }$$

*P* represents the image after contrast enhancement, *L* represents the image to be enhanced and here the de-noising image, and γ represents the correction degree. The smaller the value is, the brighter the image is. The gamma correction curve is shown in Figure 2. The value of γ is bounded by 1. For γ>1, the high light part will be suppressed the low light part will be expanded, and the image will become dark as a whole. For γ<1, the high light part is extended, the low light part is suppressed, and the whole image becomes bright.

The results of gamma correction make the image brighter or darker as a whole. Our goal is to obtain an image with enhanced target and clear contrast. Therefore, this paper proposes an improved method of weighted local mean to improve the image contrast. The specific methods are as follows: (2)$$P=L^{\mu +0.5}$$
(3)$$\mu \left(i,j\right)=\alpha \times \left(\beta \times L\left(i,j\right)+\left(1-\beta \right)\times H\left(i,j\right)\right)$$
(4)$$H\left(i,j\right)=\frac{1}{r\times r}\sum_{m=-(r-1)/2}^{(r-1)/2}\sum_{n=-(r-1)/2}^{(r-1)/2}L^{2}\left(i+m,j+n\right)$$
where *r* is the size of the local window; H(i,j) is the local mean; α and β are the coefficients that control the tensile degree.

The algorithm proposed in this paper is compared with histogram equalization, gamma correction, LIP and high-frequency enhanced filtering (HEF) algorithms. The contrast effect is shown in the Figure 3 below. The contrast areas are marked with rectangular boxes:

From the comparison effect of local regions, the method in this paper has a good effect. HE direct use of image histogram to adjust the contrast is simple and effective, but this method in the image useful data contrast is very close when the effect is better, and the experimental image useful data contrast is not all close, so the effect is general; the LIP method can expand the low gray value part of the image, display more details of the low gray value part, compress the high gray value part, reduce the details of the high gray value part, so as to achieve the purpose of emphasizing the low gray part of the image, and enhance the sky part from the effect, which leads to the contrast between the sky and the tree. HEF contrast enhancement is obvious, but the red frame-out area weakens the highlight effect of the light and leads to excessive unnaturalness at the junction of the tree and the sky. Gamma correction corrects the image from the whole, and the overall effect is good, but the correction of the local area needs to be improved. The local effect of the improved method in this paper is obvious.

In addition, since the source image inevitably contains some effects such as noise, we added the method of robust principal component analysis (RPCA) to remove the noise of small structures [21]. Robust principal component analysis can effectively find out the ‘main’ elements and structures in the data, which is different from the principal component analysis. RPCA believes that the data matrix can be decomposed into two parts, the low-rank part and the sparse part, and the noise is sparse. Therefore, the de-noising effect can be achieved by retaining the low-rank part. RPCA principle formula is as Equation (Equation 5): (5)I=L+S
where *I* represents the source image, and *L* and *S* represent the low-rank and sparse parts after RPCA decomposition.

### 2.2. Visual Saliency Map Extraction

Through the observation of a large number of infrared images, it is found that due to the unique imaging characteristics of infrared images, the appearance of infrared images on the target is particularly prominent under the conditions of weak light and smoke occlusion. Especially under the conditions of smoke, the target in the visible light image may be hidden. Therefore, it is particularly important to guide the fusion image through the infrared target saliency map.

Zhai et al. proposed a saliency detection method [22], in which the saliency value of each pixel is the sum of the Euclidean distance between the brightness value of the pixel and the brightness value of other pixels. Assuming that $I_{p}$ represents the brightness value of pixel *p* in image *I*, the formula is as Equation (Equation 6).
(6)$$S\left(p\right)=\left|I_{p}-I_{1}\right|+\left|I_{p}-I_{2}\right|+\ldots +\left|I_{p}-I_{N}\right|$$
where *N* represents the total number of pixels in image *I*. The algorithm is simple and effective. In this paper, the algorithm is introduced to extract the image saliency value, which is used to guide the weight value of the fused image saliency target and the weight coefficient of the fusion infrared image and visible image in the basic layer. Before extracting the saliency map, we add the filtering operation to avoid the influence of noise points. The formula is as follows: (7)$$I_{SG}=Gaussian\left(P\right)$$
(8)$$S=\sum_{j=0}^{C-1}M_{j}\left|I_{SG}-I_{j}\right|$$
where *P* represents the enhanced infrared image, $I_{SG}$ the image after Gaussian filtering, *j* the pixel intensity, $M_{j}$ the number of pixel intensity values, *C* the gray level being set to 256 in this paper and $I_{j}$ the brightness value *j*.

### 2.3. Multi-Scale Decomposition Based on Improved Differential Rolling Guided Filter (DRGF)

The Gaussian filter is widely used in the field of multi-scale image processing. A Gaussian filter will filter out the noise and small structure in the source image, at the same time blurring the whole image and producing artifacts. The edge-preserving filter can retain the image boundary content and reduce artifacts. The most widely used edge-preserving filters are weighted least squares filter, bilateral filter and guided filter. The guided filter is to filter the initial image through a guided image so that the output image is generally similar to the initial image, but the texture part is similar to the guided image. The guided filter can not only realize the advantages of bilateral filtering to maintain the edge but also overcome the shortcomings of bilateral filtering in gradient deformation near the edge. Zhang et al. [23] improved the guided filtering and proposed an *RGF* technology, which has the characteristics of scale perception and edge preservation. This method includes small structure removal and edge recovery. *RGF* is composed of a Gaussian filter and guided filter, and the specific implementation is as follows: (9)$$G=Gaussian(I,\sigma_{s})$$
(10)$$J^{t+1}=GuideFilter\left(J^{t},I,\sigma_{S},\sigma_{r}^{2}\right)$$
where, $Gaussian(I,\sigma_{s})$ represents Gaussian filtering, $GuideFilter(J^{t},I,\sigma_{S},\sigma_{r}^{2})$ represents guided filtering, *I* represents the image to be filtered, $\sigma_{s}$ is used as a proportional parameter to control the Gaussian filtering removal scale. Represents the boot image, controls the distance weight, set to 0.05. *RGF* is iteratively implemented by Equations (Equation 9) and (Equation 10), which can be simply expressed as: (11)$$u=RGF(I,\sigma_{s},\sigma_{r},T)$$
where *T* is the number of iterations.

The multi-scale decomposition formula of improved DRGF is: (12)$$M^{i}=RGF\left(M^{i-1},\sigma_{s}^{i-1},\sigma_{r},T\right),i=1,\ldots ,N$$
(13)$$D^{i}=\left\{\begin{array}{c} M^{i}-P,i=1\\ M^{i-1}-M^{i},i=2,\ldots ,N \end{array}\right.$$
(14)$$B=M^{i},i=N$$
where *P* is the enhanced source image, $M^{i}$ is the filtering image of the *i* layer, $D^{i}$ is the detail layer of the *i* layer, *N* is the decomposition layer, $M^{i}$ as the basic layer, $\sigma_{s}^{i}$ and $\sigma_{r}$ is the filtering parameter, this paper sets $\sigma_{s}^{i}=2\sigma_{s}^{i-1}$ to control the image scale; *T* is the number of iterations. The multiscale decomposition structure is shown in Figure 4.

### 2.4. Fusion of Base Layers

The image base layer after multi-scale decomposition contains the basic information of the image. The traditional ‘average’ fusion rule is not sensitive to the contrast of the image, which will weaken the contrast of the image and weaken the prominent information. In Section 2.2, this paper proposes a method for extracting infrared dominant images. Equation (Equation 8) is used to calculate the dominant values of two images in the base layer, and then Equations (Equation 15) and (Equation 16) is used for image fusion in the base layer. The formula is as follows: (15)$$\omega_{B}=0.5+0.5\times \left(S_{R}-S_{V}\right)$$
(16)$$BF=\omega_{B}B_{R}+\left(1-\omega_{B}\right)B_{V}$$
where $B_{R}$ and $B_{V}$ represent the infrared image base layer and the visible image base layer, $\omega_{B}$ the fusion weight of the infrared image base layer, $S_{R}$ and $S_{V}$ the obvious indigenous value of the infrared image base layer and the visible image base layer, BF the image after the fusion of the base layer.

The “weighted average” method, the information entropy method and the regional average energy method are compared and the results are shown in Figure 5.

### 2.5. Fusion of Detail Layers

The multi-scale decomposition of the image detail layer contains a variety of scale details of the original image, and this subtle detail information is useful. Due to the different imaging principles of the two source images, the information contained in the detail layer is complementary. Therefore, the traditional fusion strategy of ‘absolute value maximization’ does not take into account the different characteristics of infrared and visible images. The fusion results may change the original image information and introduce irrelevant details and noise. The ‘additive fusion’ strategy will weaken the contrast of detail features. In order to avoid this phenomenon, we propose a method based on the combination of pixel intensity and gradient to fuse the detail layer. The pixel intensity can represent the energy information of the highlight detail part, and the gradient represents the highlight degree of the image. Combined with the advantages of the two methods, the detail highlights of the two source images can be fully retained. The specific formulas are as follows: (17)$$\omega_{D}=\left\{\begin{array}{c} 0,\left|D_{V}\right|<\left|D_{R}\right|\&G_{V}<G_{R}\\ 1,\;\mathrm{others}\; \end{array}\right.$$
(18)$$DF=\omega_{D}\times D_{V}+\left(1-\omega_{D}\right)\times D_{R}$$

In the expression, $D_{V}$ and $D_{R}$ denote the pixel values of the visible light image detail layer and infrared image detail layer, $G_{V}$ and $G_{R}$ denote the gradient values of visible light image detail layer and infrared image detail layer, $\omega_{D}$ denotes the fusion weight of visible light image detail layer, DF denotes the fusion image detail layer.

In order to prove that our method can preserve detail information more effectively, it is compared with ‘additive fusion’, ‘absolute value is larger’ and the method in Reference [17] (VSM-WLS), and the results are shown in the Figure 6.

By means of the above methods, we obtain the fused basic layer, detail layer and infrared saliency map, and then overlay these three parts. Among them, the infrared saliency map is used to guide the infrared target of the fused image, so we give it a weight to avoid losing the original information of the image.

Finally, the basic layer, the detail layer and the infrared image are weighted and added by Formula (Equation 19).
(19)$$F=BF+DF+\omega_{S}S_{R}$$
where *F* is the final fusion image, and $\omega_{S}$ is the fusion weight of the infrared dominant image.

## 3. Results and Analysis

The infrared and visible images in the experiment are four groups of images accurately registered from the TNO dataset. The image scene environment covers night, dark environment and dense fog environment. In order to verify the effectiveness of the algorithm in all aspects, this paper evaluates the fusion effect through subjective evaluation and objective evaluation. The comparison algorithms include curvelet transform (CVT) [24], DTCWT, Haar, LP, MSVD, NSCT, NSST, potential low-rank representation decomposition (LatLRR) [14], and VSM-WLS. The experimental platform is MATLAB 2018b (Core i7, clocked at 3.30 GHz, memory of 16 GB).

### 3.1. Parameter Setting

In this method, there are three adjustable parameters, namely α and β in Formula (Equation 3) and $\omega_{S}$ in Formula (Equation 19). The corresponding fine-tuning can be made according to different application requirements. Here’s how we determine the parameters:

In Formula (Equation 3), α is the overall correction parameter, and its value affects the overall brightness and darkness of the image. If the value is too small, the whole image is bright and the contrast is reduced. On the contrary, if the value is too large, the image is too dark and some details are lost. Based on experience, the α value is 1.9.

β in Formula (Equation 3) and $\omega_{S}$ in Formula (Equation 19) are both weight parameters. We introduce a nonlinear function to control the value parameter. The nonlinear function is shown in Formula (Equation 20).
(20)$$S_{\lambda }\left(\xi \right)=\frac{arctan(\lambda \xi )}{arctan\lambda }$$
where ξ is the independent variable of the function, and the range of ξ and $S_{\lambda }\left(\xi \right)$ is [0, 1]. The nonlinear transformation function under different λ values is shown in Figure 7. It can be seen that when λ gradually increases, the curve becomes steeper and steeper, and the nonlinear transformation is also gradually enhanced. Therefore, we can control the guiding weight of the infrared image by adjusting λ.

Therefore, the two weight parameters can be expressed as: (21)$$\omega_{S}=\frac{arctan(\lambda S_{R})}{arctan\lambda }$$
(22)$$\beta =\frac{arctan(\lambda_{\beta }H)}{arctan\lambda_{\beta }}$$

Among them, $S_{R}$ represents the normalized infrared image, *H* represents the mean of local gray level of normalized image, λ and $\lambda_{\beta }$ are used to adjust the weight according to different applications. If λ is too large, it will lead to too much infrared information and lose the original image information. On the contrary, too little infrared information cannot play a guiding role, so does $\lambda_{\beta }$. The λ and $\lambda_{\beta }$ values in this article are set to 8 and 9, respectively.

### 3.2. Subjective Evaluation

The first set of data is the image of the intersection in the dark environment. The size of the source image is 575×475. The infrared image highlights the target information of pedestrians, street lights, cars and so on. The visible image can only highlight the information of billboards, street lights, signal lights and so on. The comparison results of different fusion algorithms are shown in Figure 8. The red box is selected where part of the target is amplified. The comparison results show that Haar, MSVD, NSST and LatLRR algorithms are too smooth, resulting in low image contrast after fusion. CVT, LP, NSCT algorithms appear artifacts around the pedestrian contour; VSM-WLS algorithm has relatively good effect, but for the prominent effect of the target it is inferior to the proposed algorithm, which can be observed from the amplified pedestrian effect.

The second set of data is the image in the dark scene, and the size of the source image is 360×270. The infrared image highlights the smoke, people and doors on the roof, and the visible image highlights the texture details of the trees, door contours and ground. The comparison results of different fusion algorithms are shown in Figure 9. Haar, MSVD, NSST, VSM-WLS algorithms have low image contrast after fusion, and CVT, LP, NSCT algorithms have artifacts on the edge of image characters after fusion, and LatLRR algorithm is not of rich information in the texture of trees and ground bricks. The proposed algorithm leads to good effect in artifacts, contrast, target, texture and so on.

The third group of data is the image of the ship on the sea in the dark scene, and the size of the source image is 505×510. The comparison results of different fusion algorithms are shown in Figure 10. From the overall contrast, Haar, MSVD, NSST and LatLRR have low contrast and unclear image. From the perspective of highlighting the target, the proposed algorithm is the best.

The fourth group of data is an image in the dense fog scenario, and the size of the source image is 620×450. In the infrared source image, the target information can be clearly displayed without the influence of dense fog. In the visible light image, the texture details of the house and land are clear, but the target is hidden. The comparison results of different fusion algorithms are shown in Figure 11. In the dense fog scenario, the highlight of the target is particularly important. The selected part of the red box in the figure is the target hidden by the dense fog after amplification. It can be seen from the comparison figure that the algorithm in this paper has the most obvious highlight effect on the target. In terms of texture detail preservation, VSM-WLS algorithm is slightly better than the algorithm in this paper for details preservation of roof. However, in terms of ground texture, the algorithm in this paper retains more details than VSM-WLS algorithm. The details of roof are selected by blue box, and the ground details are selected by green box.

### 3.3. Objective Evaluation

At present, many researchers have conducted research on the image fusion evaluation index [25,26], which is mainly divided into four types, namely based on information theory, image features, image structure similarity and human perception measurement. This paper selects several evaluation indexes to verify the objective effect of the algorithm. The selected indicators are: information entropy (EN), structural similarity (SSIM), average gradient (AG) and standard deviation (SD).

The main function of EN is to measure the amount of information in the image. The greater the value of information entropy is, the greater the amount of information contained is, and the more retained details of the image are. The main function of SSIM is to measure the structural similarity between the fused image and the two source images. The larger the value, the closer the representation and the source image are, the more details are retained. The main function of AG is to reflect the clarity of the fused image, and also to reflect the small detail contrast and texture transformation characteristics in the image. The larger the value is, the clearer the image is; the larger the SD value, the higher the image quality and the clearer the image.

The four groups of fusion images based on different algorithms are quantitative evaluated and indexed as shown in Table 1. The best indicator in the table is marked by bold fonts, and the second value of the effect is marked by underline. The Table 1 (S1–S4) represent four test images.

It can be seen from Table 1 that in terms of objective evaluation indexes, the proposed method has certain advantages in EN, SSIM, AG and SD. On the whole, most of the indexes of this algorithm are higher than those of other comparison algorithms. Although it does not rank first for EN in the second group, the SSIM and AG in the third group and the SSIM in the fourth group, it ranks second and leading. The algorithm in this paper takes into account the enhancement of image contrast. In terms of indicators, the AG and SD are basically in the leading position, especially the standard deviation index of the third group of images, which is 51.37% higher than that of the second group. By comprehensive comparison, this method performs well in objective evaluation.

Finally, the running time of different methods on four experimental images is provided in Table 2. Our method is at the intermediate level. CVT, LP and MSVD algorithms are the most basic algorithms, and the running time is relatively short. The running time of the proposed method is faster than that of NSST and LatLRR, and slightly lower than that of NSCT, but the fusion effect is better than NSCT. The algorithm in this paper is similar to the VSM-WLS algorithm, and both of them use the method of saliency map and multi-scale decomposition. However, the algorithm in this paper has been improved in the fusion algorithm of each layer of multi-scale decomposition, and many factors are considered. Therefore, the algorithm complexity is high, but the comprehensive effect is better than that of VSM-WLS. The algorithm in this paper needs to be improved in terms of time complexity, which is also the direction for further efforts.

## 4. Conclusions

This paper presents an infrared and visible image fusion method based on visual aboriginal image and image contrast enhancement. The contrast enhancement algorithm is used to improve the image contrast and the clarity of the target. The infrared salient-indigenous image is used to guide the prominence of the salient-indigenous target in the fusion image. The differential rolling guidance filtering method is used to decompose the image into different layers, so as to facilitate the targeted fusion. In terms of the fusion strategy of the basic layer and the detail layer, the fusion weights are controlled by the salient-indigenous value and the fusion method based on the combination of pixel intensity and gradient is also proposed, which is targeted for fusion. The experimental results show that the retention of the target is very prominent, and the detail information retention is better than other algorithms. Since the algorithm in this paper is comprehensive and complex, the next step is to improve the speed.

## Figures and Tables

**Figure 1 sensors-22-06390-f001:**
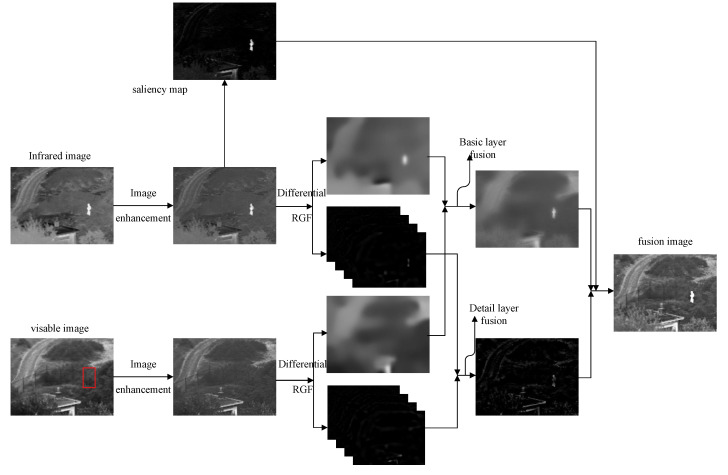
Infrared and visible image fusion frame diagram. The red rectangle frames the visible image where the person is located.

**Figure 2 sensors-22-06390-f002:**
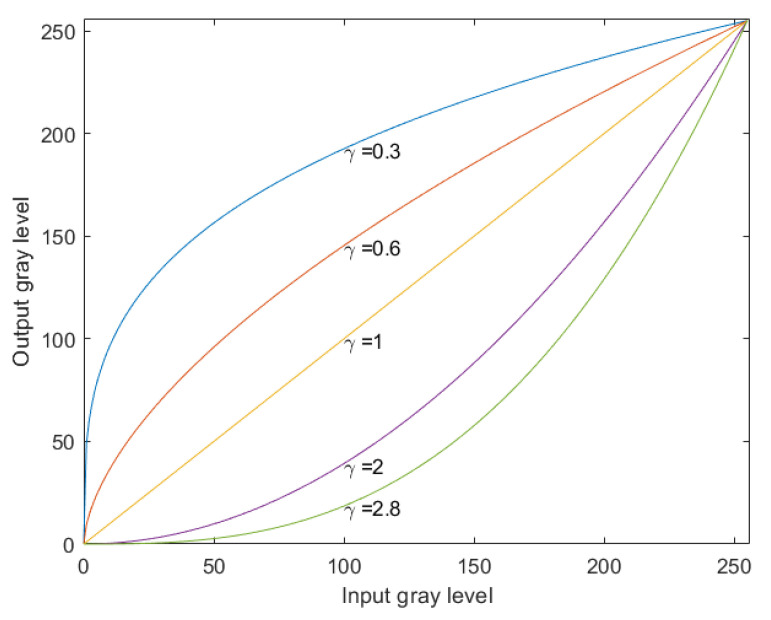
Gamma correction curve.

**Figure 3 sensors-22-06390-f003:**
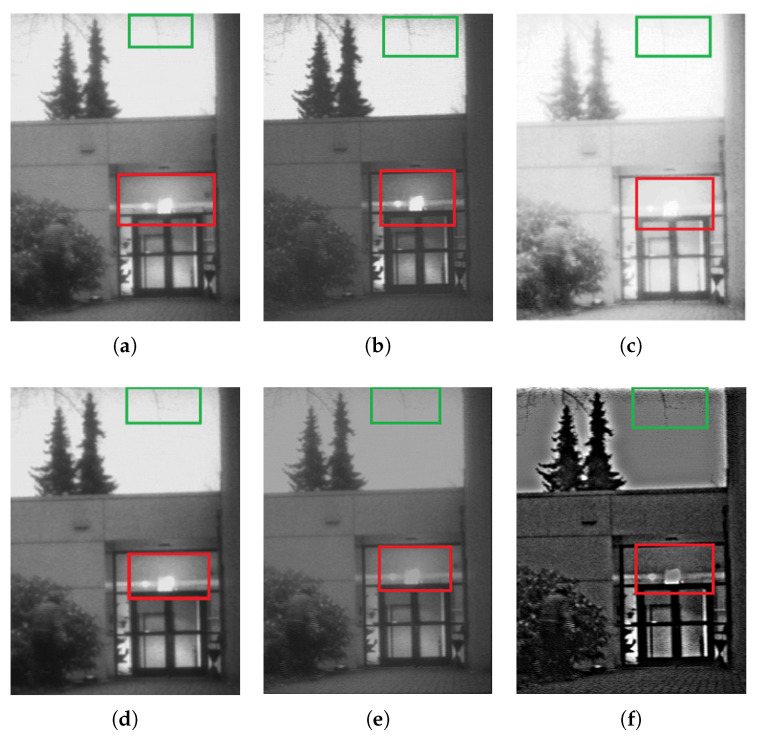
Image contrast enhancement algorithm, The red box and the green box are contrast areas: (**a**) Visible image; (**b**) Proposed; (**c**) HE; (**d**) Gamma correction; (**e**) LIP; (**f**) HEF.

**Figure 4 sensors-22-06390-f004:**
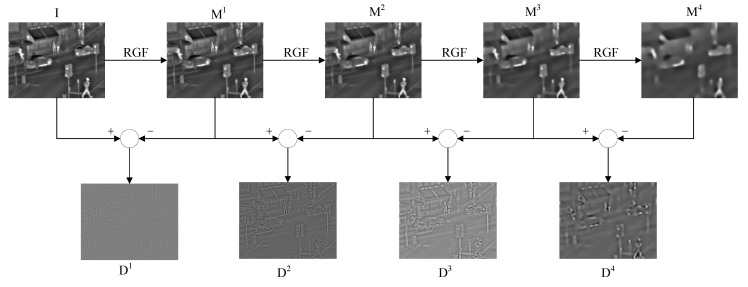
Image multi-scale decomposition process.

**Figure 5 sensors-22-06390-f005:**
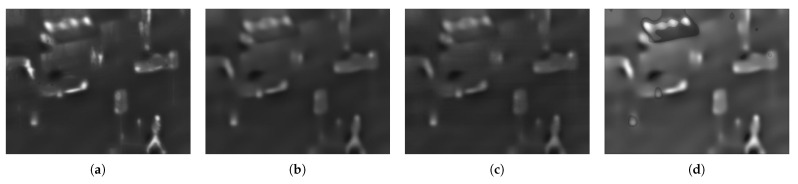
Comparison map of base layer fusion: (**a**) Proposed; (**b**) Weighted Average; (**c**) Based on Information Entropy; (**d**) Based on Average Energy.

**Figure 6 sensors-22-06390-f006:**
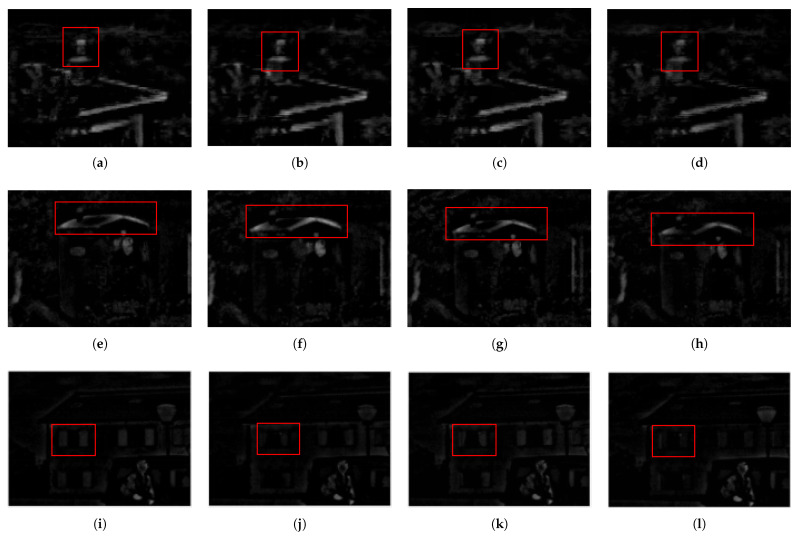
Fusion Comparison Graph, The red box section is the contrast area: (**a**,**e**,**i**) Proposed; (**b**,**f**,**j**) Additional fusion; (**c**,**g**,**k**) Absolute value is large; (**d**,**h**,**l**) VSM-WLS.

**Figure 7 sensors-22-06390-f007:**
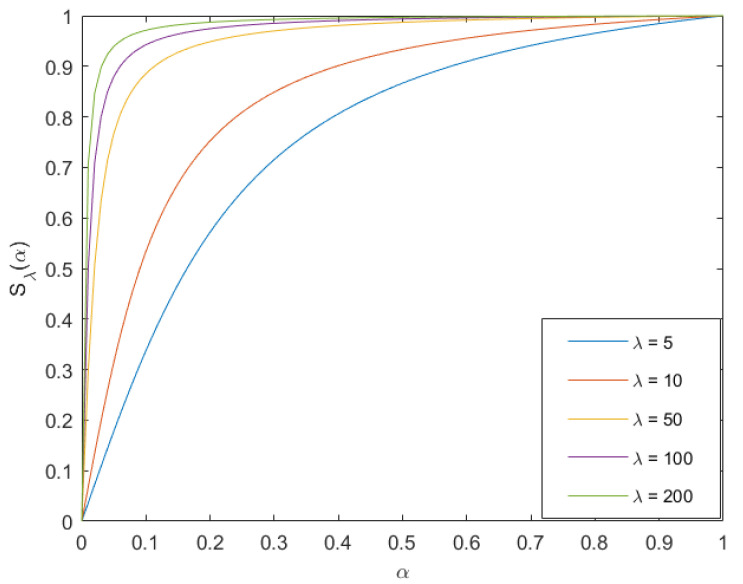
Nonlinear graph.

**Figure 8 sensors-22-06390-f008:**
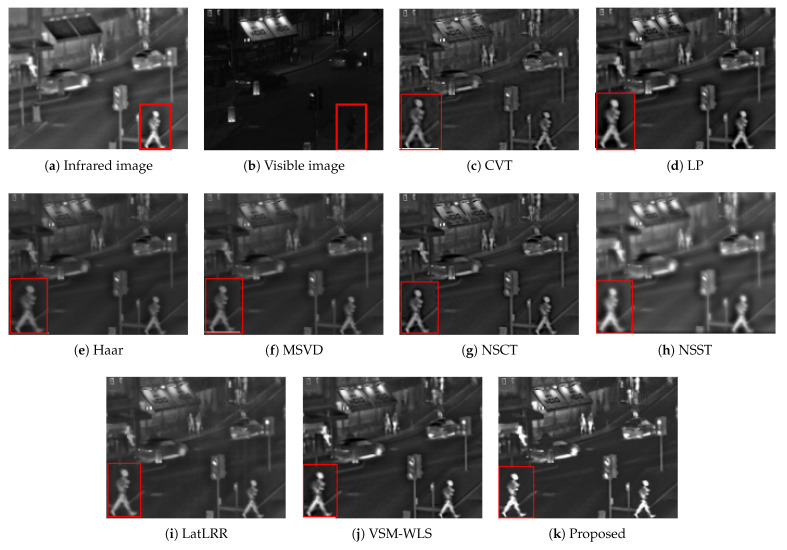
Image fusion results of the first group. The red box in the first two images is the main contrast area, and the red box in the remaining images is the enlarged effect of the contrast area.

**Figure 9 sensors-22-06390-f009:**
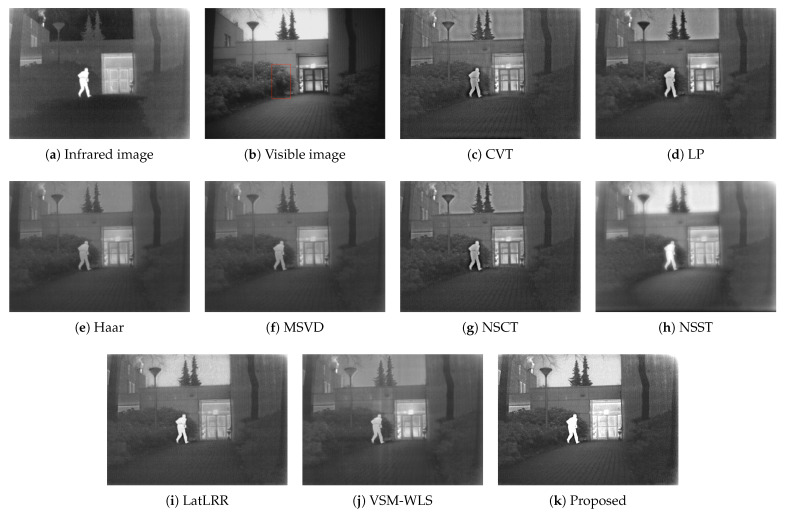
Image fusion results of the second group. The red box part is the location of the person in the visible image.

**Figure 10 sensors-22-06390-f010:**
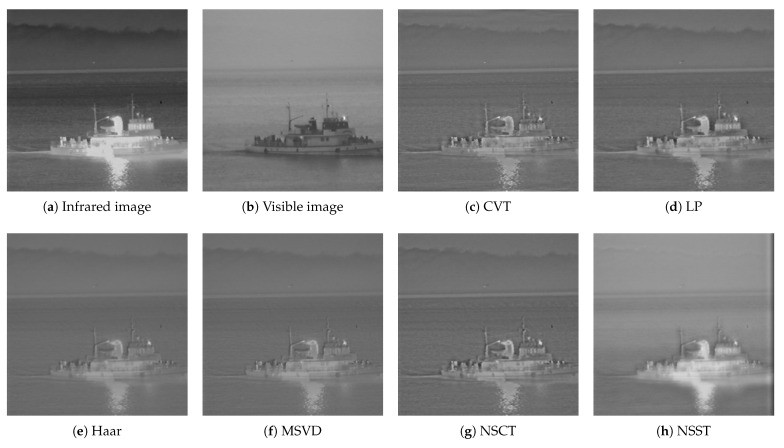
Image fusion results of the third group.

**Figure 11 sensors-22-06390-f011:**
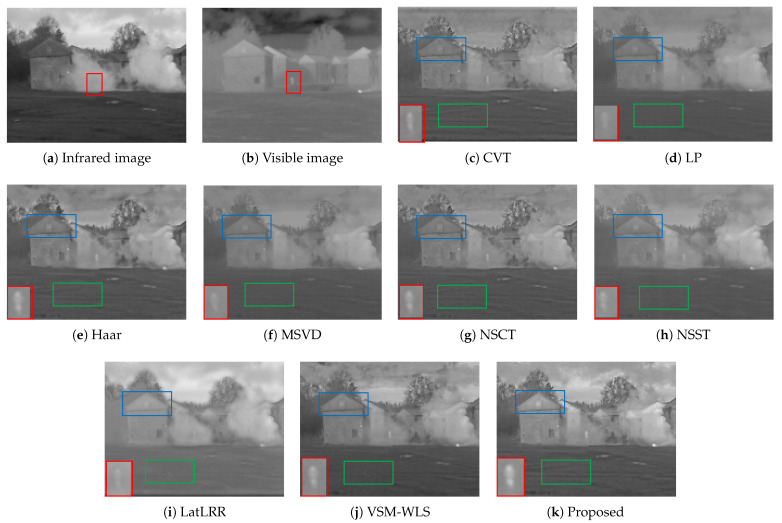
Image fusion results of the fourth group. The first two images of the red box part is the character position, the rest of the image red box part is the character amplification effect, green box and blue box part are significant contrast area.

**Table 1 sensors-22-06390-t001:** Objective evaluation results of different fusion methods.

Image	Evaluation	Methods							
		Proposed	CVT	LP	MSVD	NSCT	NSST	LatLRR	VSM-WLS
S1	EN	**6.7397**	6.1692	6.4729	5.935	6.1957	6.7243	6.0297	6.282
	SSIM	0.616	0.602	0.6165	**0.6375**	0.6199	0.603	0.6315	0.6371
	AG	**3.9535**	3.2681	3.4755	2.1762	3.3247	2.391	2.4135	3.4792
	SD	**40.3372**	25.1804	31.5615	21.1486	26.264	32.8052	27.4851	33.1943
S2	EN	7.048	6.7779	6.7763	6.5541	6.7209	**7.2385**	6.7016	6.8267
	SSIM	**0.7735**	0.7158	0.7257	0.7675	0.7328	0.7229	0.7531	0.742
	AG	**4.3621**	3.7283	3.8093	2.5071	3.7272	2.3984	2.529	3.861
	SD	**56.5402**	34.1581	36.2648	31.6308	33.8815	50.3687	35.757	44.5028
S3	EN	**5.8723**	5.2067	5.3071	4.8954	5.1643	5.8654	5.076	5.6764
	SSIM	0.8432	0.8123	0.816	**0.8481**	0.8238	0.8009	0.8412	0.7929
	AG	2.7097	2.0006	2.0467	1.2306	2.043	1.334	1.4056	**2.7673**
	SD	**30.7614**	12.4632	14.0743	10.4837	12.6503	19.481	12.8921	20.3214
S4	EN	**6.8876**	6.5601	6.492	6.2407	6.442	6.7492	6.4338	6.7884
	SSIM	0.7819	0.7596	0.7637	0.8066	0.7728	0.7577	**0.7959**	0.7688
	AG	**3.5948**	3.1057	3.1007	1.808	3.0597	1.82	1.7632	3.5846
	SD	**40.469**	29.9993	30.6485	27.6401	29.2161	37.2923	30.0467	37.0782

**Table 2 sensors-22-06390-t002:** Running Time Comparison on four experimental images.

Time (s)	Methods							
	Proposed	CVT	LP	MSVD	NSCT	NSST	LatLRR	VSM-WLS
S1	2.7117	0.9830	0.0087	0.2617	2.4595	6.4310	54.2865	1.2896
S2	2.4163	0.7733	0.0065	0.2121	2.1705	6.9734	152.0037	1.3355
S3	2.2129	0.7543	0.0051	0.1895	1.9775	4.0279	55.9909	1.0919
S4	2.6183	0.9489	0.0073	0.2296	2.3378	5.9728	163.1223	1.3334

## Data Availability

Source of data set in experimental analysis: https://figshare.com/articles/dataset/TNOImageFusionDataset/1008029.

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
