# Peer review of "Infrared and Visible Image Fusion Based on Visual Saliency Map and Image Contrast Enhancement"

_sensors, 2022, doi:10.3390/s22176390_

Round 1

Reviewer 1 Report

- The complexity of all the considered methods, including that of the proposed approach, should be included in the paper.

- It is not clear if the comparison between algorithms is fair because not all the parameters of the competing methods are not given. To facilitate reproducible research, I suggest that the authors release the related source codes on github.com, the website of the authors' research group, or a similar website. This could make a positive impact on the academic community.

- The visible low light image enhancement approach based on recursive filtering and contrast stretching techniques, driven by statistical measures of the image and implemented under a logarithmic image processing (LIP) model should be mentioned. The LIP effect is close to gamma correction in some cases and the IIR filtering resembles a one-dimensional line local mean for contrast enhancement.

- All the acronyms should be defined when first used (e.g. VSM-WLS).

- There should be proper typesetting of omega_s after Eq. 19.

Reviewer 2 Report

Author present image fusion  between infrared and visible image  based on visual visible image and image contrast enhancement. The concepts is interesting but there is some points needed to be calified  priror to the publication.

1) Image fusion should refer to fuse image taken at different modality at the same time. In figure 1 and 7, in the visible image why we can't see the person in the image.

2) Contrast improvement using gamma correction is not a suitable method. The technique should be used to correct the brightness in which gamma less than one is used to correct too dark image and vice versa  Why don't use contrast stecthing where it wil involve in modify image histogram directly. Anyway please also comment on how alpha and beta in equation (3)  are set.

3) in figure (3), we see nothing and see no differcence in details images.  

Round 2

Reviewer 1 Report

The authors have addressed my comments.

Reviewer 2 Report

I feel satisfied with author responses in the current revised version.

I have no further comments.